# Online Adaptive MRI-Guided Stereotactic Body Radiotherapy for Pancreatic and Other Intra-Abdominal Cancers

**DOI:** 10.3390/cancers15215272

**Published:** 2023-11-03

**Authors:** Danny Lee, Paul Renz, Seungjong Oh, Min-Sig Hwang, Daniel Pavord, Kyung Lim Yun, Colleen Collura, Mary McCauley, Athanasios (Tom) Colonias, Mark Trombetta, Alexander Kirichenko

**Affiliations:** 1Radiation Oncology, Allegheny Health Network, Pittsburgh, PA 15012, USA; paul.renz@ahn.org (P.R.); seungjong.oh@ahn.org (S.O.); min-sig.hwang@ahn.org (M.-S.H.); daniel.pavord@ahn.org (D.P.); kyunglim.yun@ahn.org (K.L.Y.); colleen.collura@ahn.org (C.C.); mary.mccauley@ahn.org (M.M.); mark.trombetta@ahn.org (M.T.); alexander.kirichenko@ahn.org (A.K.); 2College of Medicine, Radiologic Sciences/Drexel University, Philadelphia, PA 19129, USA

**Keywords:** MRI in RT, MRI-Linac, MRI-guided SBRT, adapt-to-position, adapt-to-shape, pancreatic cancers, abdominal cancers, online adaptive planning, Unity^®^, stereotactic body radiation treatment

## Abstract

**Simple Summary:**

MRI can provide better visualization of tumors and nearby organs at risk (OAR) than CT for fast and accurate contouring during online adaptive MRI-guided stereotactic body radiation treatment (MRI-guided SBRT) for pancreatic and other intra-abdominal cancers. Pre-set MRI sequences provided in a 1.5T MRI scanner hybrid with a linear accelerator can be used during MRI-guided SBRT, but they often limit tumor and OAR visualization and require a long image acquisition time. This study retrospectively analyzed 26 patients with pancreatic and intra-abdominal cancers that underwent CT and MR simulations and 3–5 fractionated MRI-guided SBRT. The visualization of tumors and OAR was improved with T1W imaging, which is essential for online adaptive planning and resulted in fast and accurate contouring in a shorter imaging time.

**Abstract:**

A 1.5T MRI combined with a linear accelerator (Unity^®^, Elekta; Stockholm, Sweden) is a device that shows promise in MRI-guided stereotactic body radiation treatment (SBRT). Previous studies utilized the manufacturer’s pre-set MRI sequences (i.e., T2 Weighted (T2W)), which limited the visualization of pancreatic and intra-abdominal tumors and organs at risk (OAR). Here, a T1 Weighted (T1W) sequence was utilized to improve the visualization of tumors and OAR for online adapted-to-position (ATP) and adapted-to-shape (ATS) during MRI-guided SBRT. Twenty-six patients, 19 with pancreatic and 7 with intra-abdominal cancers, underwent CT and MRI simulations for SBRT planning before being treated with multi-fractionated MRI-guided SBRT. The boundary of tumors and OAR was more clearly seen on T1W image sets, resulting in fast and accurate contouring during online ATP/ATS planning. Plan quality in 26 patients was dependent on OAR proximity to the target tumor and achieved 96 ± 5% and 92 ± 9% in gross tumor volume D_90%_ and planning target volume D_90%_. We utilized T1W imaging (about 120 s) to shorten imaging time by 67% compared to T2W imaging (about 360 s) and improve tumor visualization, minimizing target/OAR delineation uncertainty and the treatment margin for sparing OAR. The average time-consumption of MRI-guided SBRT for the first 21 patients was 55 ± 15 min for ATP and 79 ± 20 min for ATS.

## 1. Introduction

Magnetic resonance imaging (MRI) combined with a linear accelerator (MRI-Linac) [1,2,3,4] yields a technique that shows promise in MRI-guided stereotactic body radiotherapy (SBRT) [5]. MRI-Linac [6] provides superior visualization of target tumors and surrounding organs at risk (OAR) to improve delineation accuracy; MRI-guidance accounts for position, size, and shape changes during online adaptive SBRT planning [7]. Therefore, MRI-guided SBRT [8,9,10] is increasingly used for pancreatic and other intra-abdominal cancers using currently two clinically available MRI-Linacs [11].

True fast imaging with steady state precession (TrueFISP) combined with breath holding (BH) is possible on the first MRI-Linac, a 0.35T MRI combined with a linear accelerator (MRIdian^®^, ViewRay Inc., Mountain View, CA, USA) [12,13,14]. A T2-Weighted with exhalation-navigating (T2W + Nav) MRI scan with the second MRI-Linac is acquired on a 1.5T MRI combined with a linear accelerator (Unity^®^, Elekta; Stockholm, Sweden) [15]. In each fraction of MRI-guided SBRT on these MRI-Linacs, one (or more) TrueFISP or T2W + Nav image sets are acquired as daily-MRIs to account for inter-fractional changes of targets and OAR. The contours can be adjusted during online adaptive planning, and patient setup can be verified before and after beam delivery while patients are on the treatment couch.

Online adaptive planning heavily relies on the image quality of daily-MRIs, which requires superior soft-tissue contrast for fast and accurate contour adjusting of the target tumor and OAR [8,16,17,18]. An MRI scan with BH acquires a 3D volume in a short period of time (i.e., 17 s with TrueFISP + BH) [19], but pancreatic tumors can move more than 4 mm during the scan [20]. On the other hand, an MRI scan with exhalation-navigating acquires a 3D volume in a longer period of time but is dependent on the breathing period and regularity in individual patients (i.e., up to 882 s with T2W + Nav) [21]. In terms of the target motion, a gating technique, which measures real-time target motion by deformably registering fast cine images to daily-MRI, is utilized to account for the respiratory-induced target motion during beam delivery [13,19]. As an alternative, T2W with a compression belt in free-breathing significantly reduced the range of target motion [22,23,24]. However, it could still include respiratory-induced motion blurring artifacts, which leads to some degree of difficulty in adjusting with daily-MRI. Therefore, fast and precise online adaptive planning requires high-quality images from the daily-MRI with minimal or no blurring artifacts. The changing positions of the patient and their internal anatomy must also be accounted for during online adaptive planning for both adapt-to-position (ATP) and adapt-to-shape (ATS) [7,25].

Pancreatic and intra-abdominal malignancies are challenging to treat with SBRT [16,26], and require high-quality imaging with appropriate motion management [27,28]. However, most studies of online adaptive MRI-guided SBRT on clinical MRI-Lianc(s) used the pre-set MRI sequences provided by the manufacturer [11,12,13,14,15], which limited the visualization of the target and OAR on MR image sets. Furthermore, the pre-set daily-MRI required a long imaging time and had poor visualization of the target and OAR, causing difficulties in fusion, contouring [21], and fraction-to-fraction contour propagation using a rigid or deformable algorithm [29,30]. Therefore, our clinic utilized a customized T1W MRI sequence to achieve rapid imaging with superior visualization of the target and OAR and improve contouring accuracy during online adaptive planning. This study retrospectively analyzed and compared visualizing tumors and OAR on T1W image sets and pre-set T2W image sets. We also evaluated the treatment data produced in multi-fractionated MRI-guided SBRT.

## 2. Materials and Methods

In this institutional review board-approved study, multiple MR image sets were acquired from patients with pancreatic and intra-abdominal tumors before undergoing multi-fractionated online adaptive MRI-guided SBRT. All MR image sets were inspected to determine which set would be used as an MRI sequence to acquire daily-MRI for superior visualization of tumors and OAR.

### 2.1. The Workflow of CT and MRI Simulations, and MRI-Guided SBRT

Our workflow comprised 4 steps (Figure 1). First, selected patients were asked to complete the first MRI screening sheet (Figure 1a). Then, each patient underwent CT and MRI simulations (Figure 1b), where free-breathing CT (FB-CT) was used to develop a CT reference plan for online adaptive MRI-guided SBRT. Four-dimensional CT (4D-CT) image sets with an abdominal compression belt (ZiFix^TM^, Qfix, PA, USA) were used to measure tumor motion range [22]. If tumor motion was equal to or less than 0.5 cm, the patient was eligible for MRI-guided SBRT and was asked to complete the second MRI screening sheet for MRI safety before the MRI simulation [31,32]. On the same day as the CT simulation, MR image sets of T2W + Nav, T1-Weighted (T1W), and T1W + Fat Saturated (FS) were acquired using a 1.5T Unity^®^, and one of them was chosen (typically T1W) to contour the target tumor and use it as a sequence of daily-MRI. The third step was SBRT planning (Figure 1c), during which a FB-CT was rigidly registered to a chosen T1W, and target tumors and OAR were contoured on a chosen T1W and FB-CT image set, respectively. A CT reference plan (_CT_Ref) was then developed on a FB-CT for all patients. Additionally, an MRI reference plan (_MR_Ref) was developed on a T1W for the first few patients.

The fourth step was treating the patient with online adaptive MRI-guided SBRT. In each fraction of MRI-guided SBRT (Figure 1d), 3 image sets for daily-MRI were acquired: 1 set for online adaptive planning (plan-MRI) and 2 sets for patient setup verifications before (verification-MRI) and after beam delivery (post-MRI). Orthogonal, sagittal, and coronal cine images were acquired to verify internal target volume (ITV) by measuring target motion range on MiM (v7.0.6, MiM Software Inc, Cleveland, OH, USA). In this study, the patient setup was maintained by using the same Unity^®^ couch top and MRI-safe or conditional immobilization devices during each CT and MRI simulation and multi-fractionated MRI-guided SBRT.

### 2.2. Patients

Patients were usually treated every other day using a 1.5T Unity^®^. The imaging, planning, and treatment data of each patient’s CT and MRI simulation and consecutive 3–5 fractionated MRI-guided SBRT were retrospectively analyzed.

### 2.3. CT and MRI Simulations with Immobilization Devices

On the same day, all patients underwent both a CT and an MRI simulation to acquire FB-CT images with and without a gadolinium-based contrast agent for OAR contouring, 4D-CT images for measuring target motion range, and multiple MR image sets of T2W + Nav, T1W, and T1W + FS for target tumor contouring. In addition, a 2D orthogonal coronal and sagittal image set (2D-Cine) was acquired to manually measure the range of target motion for individual patients and determine the margin of ITV. The target motion range was used to verify and adjust the initial target motion range measured in 4D-CT. An abdominal compression belt, which is safe for both CT and MRI scanning, was used for abdominal imaging to manage target and organ motion up to 0.5 cm in all directions induced by respiration [22]. The pressure level of an abdominal compression belt for each patient, measured in a CT simulation, was used to set up that patient in an MRI simulation and throughout the entire MRI-guided SBRT. Two patients were scanned without the abdominal compression belt due to their discomfort.

Imaging parameters of the T1W and T1W + FS sequences were optimized during scans of volunteers and the first few patients in the study. The optimized MRI sequences were then used for all patients. For the MRI simulation, we used a T2W + Nav with a 3D turbo spin echo (TSE) MRI pulse sequence and a T1W and T1W + FS of a 3D turbo field echo (TFE) MRI pulse sequence on a 1.5T Unity^®^, with 2 MRI receiver coils (a 4-channel anterior coil and a 4-channel posterior coil). Imaging parameters of T2W + Nav were repetition time (TR)/echo time (TE) = 1800/205 ms, field of view (FOV) = 400 × 400 mm^2^, pixel size = 1.56 × 1.56 mm^2^, image matrix = 480 × 480, thickness = 2.4 mm, flip angle = 90°, bandwidth = 727 Hz, and number of signals (average = 5). Each MR image set took approximately 228 to 410 s; 233 images were acquired in total. A navigating window was set at the liver dome scout (1/3 on the lung side and 2/3 on the liver side).

Imaging parameters of T1W were TR/TE = 4.5/2.2 ms, FOV = 400 × 400 mm^2^, pixel size = 1.1 × 1.1 mm^2^, image matrix = 280 × 280, thickness = 2 mm, flip angle = 10°, bandwidth = 383, Hz and number of signal average = 5. Each MR image set took approximately 120 s; 161 images were acquired in total. All image sets acquired during the CT and MRI simulations were transferred to MiM in a Digital Imaging and Communications in Medicine format.

### 2.4. SBRT Planning

CT and MRI simulation image sets were used to develop 1 or more _CT_Ref or _MR_Ref plans for each patient. For the first 5 patients in the study, we developed both a _CT_Ref and a _MR_Ref, but we developed only a _CT_Ref for the rest of the patients. The T1W image sets were used to develop _MR_Ref plans through off-line adaptive ATS planning from the _CT_Ref plans. Average electron densities of the tumor and OAR from the _CT_Ref were assigned to corresponding contours in the _MR_Ref. Both _CT_Ref and _MR_Ref plans were developed to meet the dose constraints listed in Table 1. One of the _CT_Ref and _MR_Ref plans was peer-reviewed in the department chart round at Radiation Oncology, Allegheny Health Network Cancer Center.

On the MiM software, the OAR delineated on a FB-CT image set were heart, kidneys, liver, spinal cord, duodenum, small bowel, stomach, jejunum, colon, bone, body, spleen, and extras (i.e., air, contrast, vein, and celiac). Gross tumor volume (GTV), delineated on a T1W (or T1W + FS) MR image set, was determined for pancreatic tumors, left adrenal tumors, and pancreatic lymph nodes. The GTV contour was transferred from the T1W image set to the FB-CT image set for addition to the OAR contours. A FB-CT image set with all contours was exported to the treatment planning system (TPS, MR-Linac Monaco v5.51.11, CMS; St. Louis, MO, USA), and a patient-specific SBRT plan was developed. Each SBRT plan with 15–45 Gy in 3–5 fractions (Table 1) was calculated using Monte Carlo^®^ and used 7–14 beams delivered in a step-and-shoot intensity-modulated radiation therapy (IMRT) technique with 7FFF (flattening filter-free) photons. A reference plan developed on the FB-CT was exported to the oncology system (Mosaiq v2.8.3, CMS; St. Louis, MO, USA) for a treatment schedule every other day.

A patient-specific quality assurance (QA) was performed for _CT_Ref and _MR_Ref plans and the first adaptive ATS plan (_MR_ATS) to verify the applicability and deliverability of these QA plans on the Unity^®^. This QA test was performed using an ion chamber (Exradin A1SLMR, Standard Imaging, Inc., Middleton, WI, USA) [33] and using an MRI-conditional cylindrical diode array dosimeter (ArcCHECK-MR, Sun Nuclear Corporation, Melbourne, FL, USA) [34].

### 2.5. Online Adaptive MRI-Guided SBRT

Patients were treated with 3–5 fractionated MRI-guided SBRT, using our Unity^®^, between May 2020 and May 2023. For each fraction, 1 _CT_Ref or _MR_Ref plan was chosen for online adaptive ATP or ATS MRI-guided SBRT, and 1 or more _MR_ATS(s) was added to the list of reference plans for the next online adaptive MRI-guided SBRT. Three T1W (or T1W + FS) image sets were acquired during each fraction. The first T1W image set, the plan-MRI, was used to account for inter-fraction changes of targets and OAR shape, position, and size. After image fusion between the _CT_Ref (or _MR_Ref) and plan-MRI by a physicist (or a therapist), an attending physician determined ATP or ATS for further online adaptive planning. Then, GTV and OAR contours, rigidly or deformably transferred from the _CT_Ref (or _MR_Ref), were adjusted to match the plan-MRI. For ATS, GTV and the contours of the stomach, duodenum, small bowel, colon, and jejunum were usually adjusted or re-delineated by an attending physician, and the contours of the air and body were adjusted or removed by an attending physicist.

Next, 2 T1W image sets, the verification-MRI and post-MRI images, were acquired before and after the beam delivery to verify patient setup. Once the patient setup was verified, the target motion, moving within the planning target volume (PTV), was visually evaluated by the attending physician prior to the beam delivery. If there were patient setup differences between the plan-MRI and verification-MRI in GTV and OAR contours, the verification-MRI was used as the new plan-MRI to repeat online adaptive planning. A little over 10% of the fractions in all patients enrolled in this study experienced the changes in anatomical position and size found in the verification-MRI.

### 2.6. Statistical Analysis of Plan Quality

For each patient, we compared the image quality of T1W, FB-CT, and T2W + Nav image sets by inspecting tumor and OAR visualization on the plan-MRI acquired during 3–5 fractionated MRI-guided SBRT. The changes in tumor and OAR contours adjusted during ATS were quantified as a function of their variability in volume. Next, the quality of online adaptive SBRT plans was evaluated by quantifying the coverage of radiation dose to tumors and OAR. Both image quality and plan quality were evaluated for all patients using MiM. Lastly, the time-consumption of each step in our workflow was analyzed in individual, online adaptive ATP/ATS plans to determine our workflow’s efficiency.

## 3. Results

All patients successfully completed CT and MRI simulations to acquire planning image sets and were treated with 3–5 fractionated online adaptive MRI-guided SBRT.

### 3.1. Patients

Twenty-six patients with pancreatic (n = 19), left adrenal (n = 3), and lymph node cancer (n = 4), who were treated using Unity^®^ between May 2021 and May 2023, were included in this study. Of the 26 patients, 16 were male and 10 were female, and the cohort had an average age of 71 years [range: 57–95] (Table 2). Average tumor volumes of reference plans (_CT_Ref or _MR_Ref) and adaptive ATP/ATS plans were measured at 37.1 mL [range: 3.4 mL to 105.5 mL] and 35.4 mL [range: 3.4 mL to 106.4 mL], respectively. The same air-pressure of an abdominal compression belt recorded at the CT simulation was reproduced during the MRI simulation and consecutive 3–5 fractionated MRI-guided SBRT.

### 3.2. CT and MR Image Sets and Target Contouring

Figure 2 shows an example of pancreatic tumor and OAR visualization on FB-CT, T2W + Nav, and T1W image sets. Like FB-CT, the boundaries of a pancreatic tumor and OAR were well visualized on T1W, but not on T2W + Nav. Regarding tumor and OAR contouring, T1W images more clearly showed the boundaries of tumors in all 26 patients than the T2W + Nav images (Figure 3). The clear interface of organs required for higher accuracy and precision of SBRT planning was shown in T1W images, but it was unclear in T2W + Nav images.

The average GTV and PTV, measured in the _CT_Ref or _MR_Ref across all 26 patients, were 36.6 mL (range: 3.4 mL to 106.4 mL) and 74.9 mL (range: 7.1 mL to 181.4 mL), respectively. A target motion of 0.2–0.4 cm was measured with 4D-CT or 2D-Cine and added to the GTV as a margin along all directions for ITV, accounting for respiratory-induced target motion when using the abdominal compression belt.

### 3.3. Plan Quality of Reference SBRT Plans

Figure 4 shows 2 examples of patient _CT_Ref and _MR_Ref plans. All QA testing _CT_Ref, _MR_Ref, and _MR_ATS plans showed a >95% passing rate for the 3%-3mm gamma analysis and a <2% point dose difference between data measured by an ion chamber and data calculated by TPS.

### 3.4. Online Adaptive MRI-Guided SBRT

Superior visualization of tumors and OAR on T1W image sets contributed to minimizing contouring uncertainty and improving the efficiency of image fusion between a chosen reference plan and plan-MRI. Furthermore, the total acquisition time of plan-MRI, verification-MRI, and post-MRI was reduced three times from 360 s for T2W + Nav to 120 s for T1W (a 67% reduction of the total acquisition time). Figure 5 shows the visualization of tumors and OAR on 5 _MR_ATS plans created with 5 fractioned online adaptive plans.

The difference in GTV and PTV contours in all _MR_ATS(s), compared to _CT_Ref (or _MR_Ref), was minimal. The average GTV was 37.1 mL in all _CT_Ref(s) and _MR_Ref and 35.4 mL in all _MR_ATS(s). Similarly, the average PTV was 74.1 mL in all _CT_Ref(s) and _MR_Ref and 72.1 mL in all _MR_ATS(s). In addition, the dose coverages of GTV D_95%_ and D_90%_ for all _CT_Ref(s) were 94.2% [range: 78.8% to 100.0%] and 96.9% [range: 86.1% to 100.0%], respectively. The dose coverages of PTV D_95%_ and D_90%_ were slightly lower at 89.7% [range: 69.3% to 99.7%] and 94% [range: 78.8% to 100%], respectively. For all _MR_ATS(s), GTV/PTV D_95%_ and D_90%_ were approximately at 83.7% [range: 16.8% to 100.0%]/91.6% [range: 52.8% to 100.0%], respectively. The dose coverages of GTV and PTV were mainly dependent on the locations of OAR, such as duodenum or jejunum (n = 10), stomach (n = 7), small bowel or colon (n = 9), and spleen (n = 11). Other OAR, such as the liver, kidneys, and spinal cords, had a negligible effect on GTV and PTV dose coverages since the beams’ gantry angles avoided these OAR.

### 3.5. Overall Time-Consumption of Online MRI-Guided SBRT in 10 Steps

The average time-consumption of online MRI-guided SBRT for the first 21 patients was 55 min for ATP and 79 min for ATS (Table 3). The most time-consuming steps were fusion/contouring and beam delivery (5 Gy to 9 Gy in each fraction with 7–13 beams), followed by patient setup, plan optimization, and plan QA/approval. 

## 4. Discussion

SBRT planning requires that the tumor be superiorly visualized in CT or MR image sets to determine the radiation dose limit for OAR in patients with pancreatic and intra-abdominal caners [5,9,12,13,14]. In this study, we utilized T1W imaging (about 120 s) to reduce the imaging time of plan-MRI, verification-MRI, and post-MRI by approximately 67%, compared to T2W + Nav imaging (about 360 s). T1W imaging also improved tumor visualization to (1) minimize delineation uncertainty, (2) reduce GTV, and (3) spare OAR during MRI-guided SBRT. We demonstrated the efficiency of T1W imaging by inspecting all images within the clear boundaries of contoured tumors and OAR.

Intra- and inter-fractional changes in a tumor’s anatomical position and shape and a patient’s setup are often found in image-guided radiotherapy, required re-planning, or online adaptive planning [35,36,37]. To account for an inter-fractional change between a reference plan (_CT_Ref, _MR_Ref, or _MR_ATS) and the plan-MRI, we performed online adaptive ATP or ATS planning, and we repeated adaptive planning within a fraction if we found an intra-fractional change between the plan-MRI and verification-MRI. To repeat adaptive planning, the latest verification-MRI was used as the plan-MRI and fused to a reference plan or the latest _MR_ATS, followed by contouring and/or plan optimization. More than one adaptive planning was required in 10% of all fractions, which could be improved by increasing patient comfort [38]. For example, we minimized the time patients stayed on the treatment couch during online adaptive planning. T1W imaging can achieve fast imaging and efficient contouring of tumors and OAR, but it still requires patient setup for those with pre-existing health conditions, causing discomfort and pain (i.e., surgery, injury, and claustrophobia), eliminating (or minimizing) the idle waiting time for the attending physician, and reducing the delivery time of uncomplicated SBRT plans [39,40,41].

The tumor contours delineated on T1W image sets matched the contours in FB-CT image sets because they account for the respiratory-induced motion during image acquisition. However, T2W + Nav image sets were acquired at exhalation in Unity^®^, so the contour size of tumors was slightly smaller than it was in T1W image sets [42]. Our ITV was determined at every fraction by encompassing GTV delineated on a T1W image set and motion range measured on orthogonal, sagittal, and coronal cine images in between inhalation and exhalation over the multiple breathing cycles (i.e., about 5–7 breathing cycles). This helps to clarify fractional breathing patterns for increasing the reliability of ITV margins in the presence of inter-fractional breathing variability [43].

Patient immobilization is critical for maintaining patient setup and preventing changes in target/OAR position and shape [32,44]. A thin or medium-sized vac-bag was tested with the first 5 patients during patient setup, but this required a long setup time. The vac-bag also required the patient to rotate so that the ATS could be achieved during the long treatment time. Instead of a vac-bag, we used an abdominal compression belt to control the target motion moving within 5 mm [22]. The setup consistency of an abdominal compression belt was dependent on the level of experience of individual therapists, and it was more consistent when the same therapist set it up every time across all simulations and multi-fractionated MRI-guided SBRT.

Our study has limitations. Limitations of this study include the basic analysis of image and plan quality using T1W imaging. However, our clinical protocol is continuously being improved to increase workflow efficiency using immobilization devices, MRI sequences, and auto-contouring and planning. Compared to our T1W sequence, the newly released MRI sequence (b3DVaneXD) and research sequences (compressed sensing and mDIXON) were not tested to compare tumor and OAR visualization. The present manual measurement of ITV margin may not be required when using a respiratory gating technique in the near future.

This is an ongoing project at our institution and will include more quantitative analysis when compared with other MRI sequences, such as b3DVaneXD, compressed sensing, and mDIXON, to provide alternative imaging for less motion-dependent imaging, faster imaging, and fat/water suppression imaging, respectively. In addition, we will assess the complexity and dose coverage of our SBRT plans by comparing them with other plans using other imaging techniques for cross-validation.

## 5. Conclusions

This was the first study that utilized a customized and optimized T1W sequence to improve the visualization of tumors and OAR for reference planning and online adaptive planning on a 1.5T Unity^®^. The tumor and OAR boundaries were clearly visible for delineation. Our results can facilitate consistent visualization of pancreatic and other intra-abdominal tumors to achieve fast and accurate MRI-guided SBRT.

## Figures and Tables

**Figure 1 cancers-15-05272-f001:**
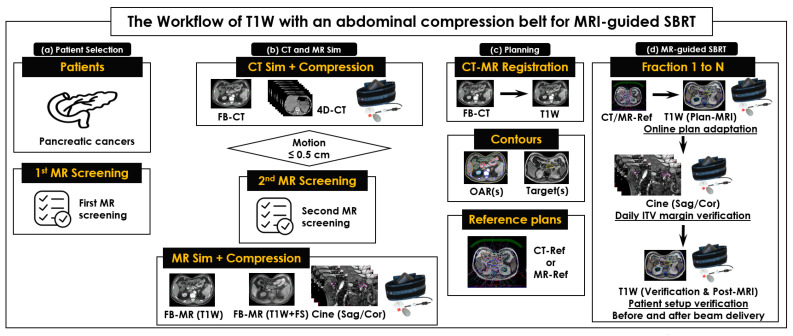
The workflow of simulations and multi-fractionated stereotactic body radiotherapy for pancreatic and intra-abdominal cancers (MRI-guided SBRT) on a 1.5T Unity^®^. (**a**) Patient selection and 1st MRI screening to check the eligibility of each patient; (**b**) CT and MRI simulations with an abdominal compression belt to acquire CT and MR image sets; (**c**) contouring organs at risk (OAR) on FB-CT images and target tumors on T1W images to develop a CT reference plan (_CT_Ref) and an MRI reference plan (_MR_Ref); and (**d**) MRI-guided SBRT in 3–5 fractions. An identical Unity^®^ couch top and the same abdominal compression belt were used in CT and MRI simulations and across all SBRT fractions. Orthogonal sagittal and coronal 2D cine images were acquired to measure the range of target motion induced by respiration for determining an internal target volume (ITV) margin.

**Figure 2 cancers-15-05272-f002:**
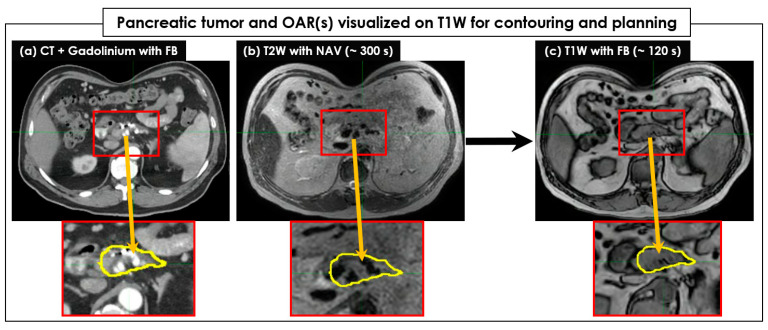
The pancreatic tumor and organs at risk (OAR) visualized on T1W for contouring and further SBRT planning. A CT image set (**a**) acquired in free-breathing (FB) with Gadolinium was compared to a T2W with exhalation-navigating (+Nav) image set (**b**) and a T1W image set (**c**). The target tumor areas were magnified, and the target tumors were contoured on a CT image set with the yellow color and copied to MR image sets. The boundary of the target tumor was clearly visible on the T1W image set acquired in a short period of time, but it was unclear on the T2W + Nav image set. The body shape due to the use of an abdominal compression belt was identical across all FB-CT, T2W + Nav, and T1W image sets.

**Figure 3 cancers-15-05272-f003:**
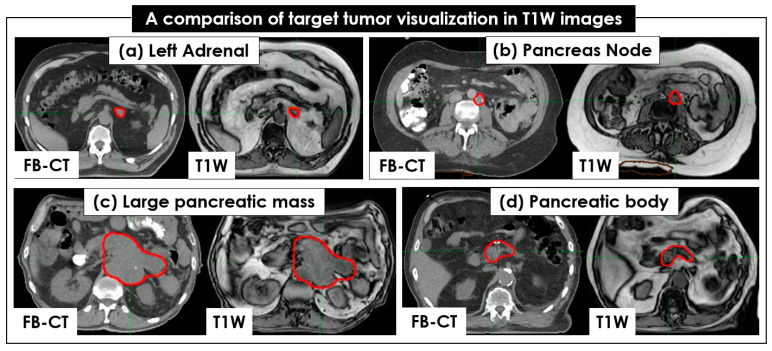
Comparison of tumor visualization between CT and 4 T1W image sets. The same target tumors of 4 patients, colored in red, are shown in free-breathing (FB)-CT and T1W: (**a**) left adrenal, (**b**) pancreatic node, (**c**) large pancreatic mass, and (**d**) pancreatic body. The target tumors are clearly seen in both the FB-CT and T1W image sets.

**Figure 4 cancers-15-05272-f004:**
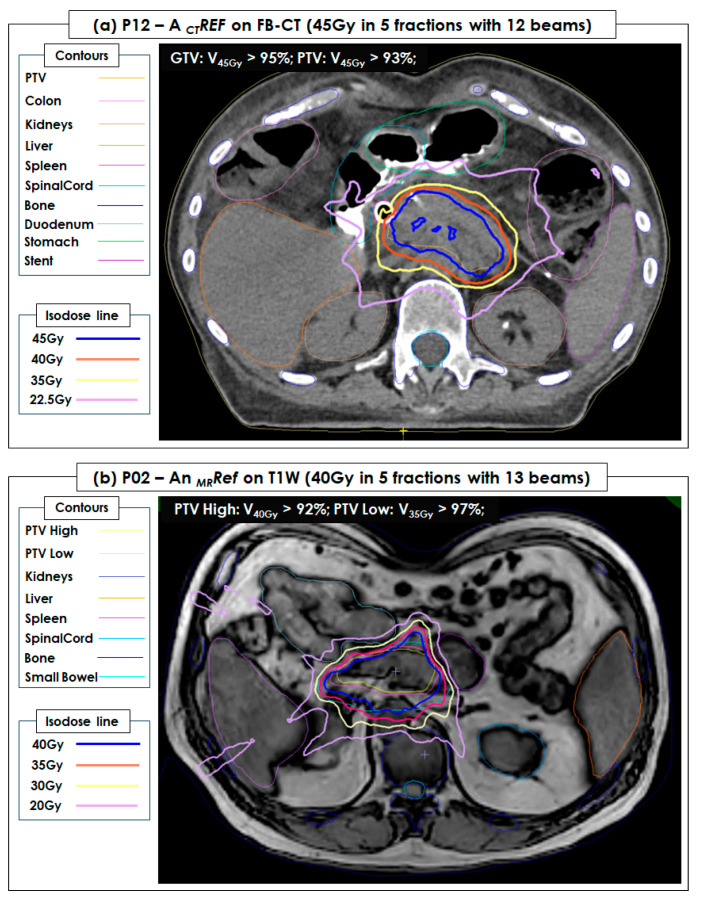
A CT reference plan (_CT_Ref) of P12 and an MRI reference plan (_MR_Ref) of P02 developed in a free-breathing (FB) CT image set (45 Gy = 9 Gy × 5 fractions with 12 beams) and a T1W image set (40 Gy = 8 Gy × 5 fractions with 13 beams), respectively. PTV and OAR (kidneys, liver, spleen, spinal cord, duodenum, bone, body (external), and small bowel) were contoured and shown in the FB-CT and T1W image sets. Both were successfully used for pancreatic and intra-abdominal cancers during 5 fractionated MRI-guided stereotactic body radiotherapy (SBRT). (**a**) The dose coverage of gross tumor volume (GTV) and planning target volume (PTV) in _CT_Ref achieved V_45Gy_ > 95% and V_45Gy_ > 93%, respectively. (**b**) The dose coverage of PTV High and PTV Low in _MR_Ref achieved V_40Gy_ > 92% and V_35Gy_ > 97%, respectively.

**Figure 5 cancers-15-05272-f005:**
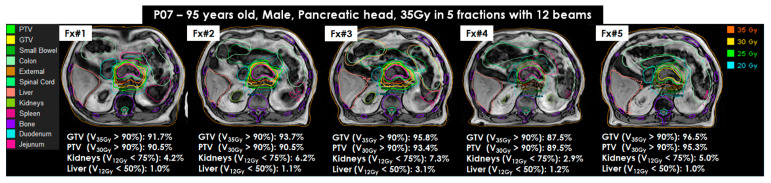
Online adaptive MRI-guided stereotactic body radiotherapy for pancreatic and intra-abdominal cancers of (MRI-guided SBRT) for P07 using 35Gy in 5 fractions with 12 beams (95 years old, male, pancreas head). All MRI adapt-to-shape (_MR_ATS) plans were created on T1W image sets. The target coverages of gross tumor volume (GTV) and planning target volume (PTV) were greater than 90% in 4 fractions (Fx#1, Fx#2, Fx#3, and Fx#5), except for Fx#4. Organs at risk (OAR), the duodenum and jejunum, were very close to the target tumor and resulted in less than 90% in Fx#4. All OAR coverages were achieved in all _MR_ATS(s).

**Table 1 cancers-15-05272-t001:** The dose constraints of planning target volume (PTV) and organs at risk for stereotactic body radiotherapy for pancreatic and other intra-abdominal cancers. This is an example of 40 Gy in 5 fractions, so the dose constraints varied on the prescription dose (15 Gy to 45 Gy).

Organs	Dose Constraints (5 Fractions) of 40 Gy in 8 Gy × 5 Fractions
PTV (or PTV_eval)	>90% coverage
GTV	90–95% Rx to cover 90–95%
Cord	V_20Gy_ < 0.03 cc
Liver	V_12Gy_ < 50%
Bilat kidneys	V_12Gy_ < 50%
Stomach PRV (2 mm)	V_40Gy_ < 0.5 cc, V_35Gy_ < 1 cc, V_30Gy_ < 2 cc
Duodenum PRV (2 mm)	V_40Gy_ < 0.5 cc, V_35Gy_ < 1 cc, V_30Gy_ < 2 cc
Small bowel PRV (2 mm)	V_40Gy_ < 0.5 cc, V_35Gy_ < 1 cc, V_30Gy_ < 2 cc
Colon PRV (2 mm)	V_40Gy_ < 0.5 cc, V_35Gy_ < 1 cc, V_30Gy_ < 2 cc
Jejunum PRV (2 mm)	V_40Gy_ < 0.5 cc, V_35Gy_ < 1 cc, V_30Gy_ < 2 cc
Spleen	< 4 Gy
Heart	Dmax ≤ 20 Gy

GTV: Gross tumor volume; PRV: Planning of organ at-risk volume.

**Table 2 cancers-15-05272-t002:** Demographic and disease profiles of 26 patients with pancreatic (n = 19) and intra-abdominal cancers (n = 7). The cohort had an average age of 71 years [range: 57–95]; 16 patients were male and 10 were female. All patients were treated with 126 fractionated MRI-guided SBRT (ATP (n = 49) and ATS (n = 77)) with the SBRT prescription (25 Gy to 45 Gy in 3–5 fractions for 25 patients and 15 Gy in 5 fractions for 1 patient). Patient P09 was treated with the stereotactic boost for the postoperative recurrence of pancreatic cancer after 45 Gy conventional fractionation.

Patient #	Diagnosis	Age	Gender	SBRT Prescription	# of Beams	Type of Adaptive Planning
P01	Pancreatic head	70	M	35 Gy in 5 fractions	13	ATP (n = 0), ATS (n = 5)
P02	Pancreatic head	65	M	40 Gy in 5 fractions	13	ATP (n = 0), ATS (n = 5)
P03	Left adrenal	61	M	30 Gy in 3 fractions	11	ATP (n = 0), ATS (n = 3)
P04	Left pancreatic lymph nodes	72	M	45 Gy in 5 fractions	13	ATP (n = 0), ATS (n = 5)
P05	Left pancreatic lymph nodes	60	F	40 Gy in 5 fractions	8	ATP (n = 0), ATS (n = 5)
P06	Left adrenal gland	63	M	30 Gy in 3 fractions	9	ATP (n = 0), ATS (n = 3)
P07	Pancreatic head	95	M	35 Gy in 5 fractions	12	ATP (n = 0), ATS (n = 5)
P08	Pancreatic head	65	F	45 Gy in 5 fractions	12	ATP (n = 4), ATS (n = 1)
P09	Pancreas Boost	64	F	15 Gy in 5 fractions	12	ATP (n = 0), ATS (n = 5)
P10	Pancreatic tail	67	M	40 Gy in 5 fractions	12	ATP (n = 3), ATS (n = 2)
P11	Pancreas	79	M	35 Gy in 5 fractions	8	ATP (n = 0), ATS (n = 5)
P12	Pancreas head/body	83	M	35 Gy in 5 fractions	11	ATP (n = 0), ATS (n = 5)
P13	Aortocaval lymph nodes	64	M	25 Gy in 5 fractions	11	ATP (n = 4), ATS (n = 1)
P14	Pancreatic body	77	F	45 Gy in 5 fractions	12	ATP (n = 1), ATS (n = 4)
P15	Pancreatic head	57	F	37.5 Gy in 5 fractions	12	ATP (n = 3), ATS (n = 2)
P16	Pancreatic head	67	F	37.5 Gy in 5 fractions	14	ATP (n = 3), ATS (n = 2)
P17	Pancreatic head	74	M	40 Gy in 5 fractions	11	ATP (n = 3), ATS (n = 2)
P18	Pancreatic head	73	F	45 Gy in 5 fractions	7	ATP (n = 4), ATS (n = 1)
P19	Portocaval node	62	M	40 Gy in 5 fractions	10	ATP (n = 5), ATS (n = 0)
P20	Pancreas	83	M	40 Gy in 5 fractions	13	ATP (n = 2), ATS (n = 3)
P21	Pancreas	71	M	45 Gy in 5 fractions	10	ATP (n = 3), ATS (n = 2)
P22	Pancreatic tail	72	F	40 Gy in 5 fractions	12	ATP (n = 2), ATS (n = 3)
P23	Pancreatic body mass	79	F	40 Gy in 5 fractions	12	ATP (n = 1), ATS (n = 4)
P24	Pancreatic head	61	F	40 Gy in 5 fractions	8	ATP (n = 4), ATS (n = 1)
P25	Pancreatic head	72	M	45 Gy in 5 fractions	8	ATP (n = 4), ATS (n = 1)
P26	Left adrenal	81	M	32 Gy in 4 fractions	7	ATP (n = 3), ATS (n = 2)
Mean ± STD or Total number		71 ± 9	M (n = 16), F (n = 10)	15 Gy to 45 Gy in 3–5 fractions	7–14	ATP (n = 49), ATS (n = 77)

P = Patient; M = Male; F = Female; ATP = Adapt-to-position; ATS = Adapt-to-shape; STD = Standard deviation.

**Table 3 cancers-15-05272-t003:** Time-consumption of online adaptive MRI-guided stereotactic body radiotherapy for pancreatic and intra-abdominal cancers (SBRT) in 10 steps. The step of fusion and contouring included some waiting time of attending physicians. The step of plan QA and approval included an independent dose verification and a visual inspection of the target tumor moving within the PTV contour.

Adaptive	MRI Screening	Patient Setup	Imaging Plan-MRI	Fusion/Contouring	Plan Optimization	Plan Review	Plan QA/Approval	Therapist Check	BEAM Delivery	Imaging Post-MRI	Seconds	Minutes
					Imaging Verification-MRI							
ATP	81 ± 35	493 ± 144	276 ± 99	763 ± 546	330 ± 180	264 ± 328	248 ± 113	17 ± 17	725 ± 238	97 ± 14	3296 ± 925	54.9 ± 15.4
ATS	77 ± 39	622 ± 254	238 ± 90	1629 ± 739	545 ± 545	337 ± 271	337 ± 396	26 ± 45	850 ± 261	98 ± 17	4759 ± 1243	79.3 ± 20.7

## Data Availability

The data can be shared upon request.

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
