# Peer review of "Online Adaptive MRI-Guided Stereotactic Body Radiotherapy for Pancreatic and Other Intra-Abdominal Cancers"

_cancers, 2023, doi:10.3390/cancers15215272_

Round 1

Reviewer 1 Report

Comments and Suggestions for Authors

1- In Abstract section, define all abbreviations at their first presence in the text.

2- In introduction section, more relevant literature are needed to clarify the necessity of the present work.

3- I materials and methods there is no reason why the researcher chosen 26 patients.

4-   In tables, the table legends should be provided self-explanatory in details. For example, in table 2, the unit of dose or Gy (SBRT prescription dose) can be moved to the legends of the table.

5- More discussion is needed to compare the results with other studies.

6- Minor editing of English language and grammar required.

Comments on the Quality of English Language

Minor editing of English language and grammar required.

Author Response

Response to Reviewer 1 Comments

Thank you for the review of our manuscript and this opportunity to improve the manuscript.

The original queries and requests are reproduced verbatim in black. Our responses to these comments are written in red.

Point 1: In Abstract section, define all abbreviations at their first presence in the text.

Response 1: Thank you for the review and comments. Four abbreviations have been defined in the abstract section – Stereotactic body radiation treatment (SBRT), T2 Weighted (T2W), organs at risk (OAR), and T1 Weighed (T1W).

Point 2: In introduction section, more relevant literature are needed to clarify the necessity of the present work.

Response 2: We apologize for the introduction section. The last paragraph in the introduction section has been revised to clarify the necessity of this study. “Pancreatic and intra-abdominal malignancies are challenging to treat with SBRT,16,26 and require high-quality imaging with appropriate motion management.27,28 However, most studies of online adaptive MRI-guided SBRT on clinical MRI-Lianc(s) used the pre-set MRI sequences provided by the manufacturer,11-15 which limited the visualization of the target and OAR on MR image sets. Furthermore, the pre-set daily-MRI required a long imaging time and had poor visualization of the target and OAR, causing difficulties in fusion, contouring,21 and fraction-to-fraction contour propagation using a rigid or deformable algorithm.29,30 Therefore, our clinic utilized a customized T1W MRI sequence to achieve rapid imaging with superior visualization of the target and OAR and improve contouring accuracy during online adaptive planning. This study retrospectively analyzed and compared visualizing tumors and OAR on T1W image sets and pre-set T2W image sets. Wealso evaluated the treatment data produced in multi-fractionated MRI-guided SBRT.

Point 3: I materials and methods there is no reason why the researcher chosen 26 patients.

Response 3: Twenty-six patients treated on Unity® in between May 2021 and May 2023 were included in this study. For clarity, the following sentence has been revised: “Twenty-six patients with pancreatic (n = 19), left adrenal (n = 3), and lymph node cancer (n = 4), who were treated using Unity® between May 2021 and May 2023, were included in this study.”.

Point 4: In tables, the table legends should be provided self-explanatory in details. For example, in table 2, the unit of dose or Gy (SBRT prescription dose) can be moved to the legends of the table.

Response 4: The authors thank you for your comments. The legend of table 1 has been revised: Table 1. The dose constraints of planning target volume (PTV) and organs at risk for stereotactic body radiotherapy for pancreatic and other intra-abdominal cancers. This is an example of 40 Gy in 5 fractions, so the dose constraints varied on the prescription dose (15 Gy to 45 Gy).

The legend of table 2 has been revised: Table 2. Demographic and disease profiles of 26 patients with pancreatic (n=19) and intra-abdominal cancers (n=7). The cohort had an average age of 71 years [range: 57 – 95]; 16 patients were male, and 10 were female. All patients were treated in 126 fractionated MRI-guided SBRT (ATP (n=49) and ATS (n=77)) with the SBRT prescription (25 Gy to 45 Gy in 3 – 5 fractions for 25 patients and 15 Gy in 5 fractions for 1 patient). Patient P09 was treated with the stereotactic boost for the postoperative recurrence of pancreatic cancer after 45 Gy conventional fractionation.

Point 5: More discussion is needed to compare the results with other studies.

Response 5: The authors agree with your comment that we need to compare the results with other studies. This initial study analyzed the current treatment data we have at our institute, but it will be expanded to analyze multi-institutional treatment data in the future. The consortium of Elekta Unity® presently gathers those data, and our institute participates in data analysis. Thus, those comparisons are being prepared for future publications.

Point 6: Minor editing of English language and grammar required.

Response 6: We apologize for the unclear language in the manuscript and have revised the manuscript to improve clarity. Three publication support writers at Allegheny Health Network have revised the grammatical mistakes in this manuscript.

Reviewer 2 Report

Comments and Suggestions for Authors

In summary, this study investigated the use of a 1.5T MRI combined with a linear accelerator for MRI-guided SBRT in pancreatic and intra-abdominal cancers. Previous attempts with preset T2W MRI sequences had limitations in visualizing tumors and nearby organs at risk (OAR). Instead, the study employed a T1W sequence, which significantly improved the visualization of tumors and OAR during online adaptive planning. This enhancement allowed for faster and more accurate contouring in a shorter imaging time. The text is generally clear and uses technical terminology appropriate for its context. However, there are some lengthy sentences that could be broken down for better readability. The main limitation of the study is the small number of patients (26).  Other than the extremely limited number of patients, the study is interesting, well-presented, and comes with appropriate results and discussions.

Author Response

Response to Reviewer 2 Comments

Thank you for the review of our manuscript and this opportunity to improve the manuscript.

The original queries and requests are reproduced verbatim in black. Our responses to these comments are written in red.

In summary, this study investigated the use of a 1.5T MRI combined with a linear accelerator for MRI-guided SBRT in pancreatic and intra-abdominal cancers. Previous attempts with preset T2W MRI sequences had limitations in visualizing tumors and nearby organs at risk (OAR). Instead, the study employed a T1W sequence, which significantly improved the visualization of tumors and OAR during online adaptive planning. This enhancement allowed for faster and more accurate contouring in a shorter imaging time. The text is generally clear and uses technical terminology appropriate for its context. However, there are some lengthy sentences that could be broken down for better readability. The main limitation of the study is the small number of patients (26). Other than the extremely limited number of patients, the study is interesting, well-presented, and comes with appropriate results and discussions.

Thank you for the review and comments.

Point 1: The main limitation of the study is the small number of patients (26).

Response 1: Twenty-six patients, treated on Unity® in between May 2021 and May 2023, were included in this study. We continuously treat more patients, so we have a gradually increasing number of patients to include in future analyses.
